

# Impacts of the Low-Level Jet's Negative Wind Shear on the Wind Turbine

Walter Gutierrez[1], Arquimedes Ruiz-Columbie[2], Murat Tutkun[3,4], and Luciano Castillo[1]

[1]Department of Mechanical Engineering, Texas Tech University, Lubbock, Texas 79409, USA
[2]National Wind Institute, Texas Tech University, Lubbock, Texas 79409, USA
[3]Institute for Energy Technology (IFE), Kjeller, Norway
[4]University of Oslo, Department of Mathematics, Oslo, Norway

*Correspondence to:* Walter Gutierrez (walter.gutierrez@ttu.edu)

**Abstract.** Nocturnal Low Level Jets (LLJs) are defined as relative maxima in the vertical profile of the horizontal wind speed at the top of the stable boundary layer. Such peaks constitute major power resources for wind turbines. However, a wind speed maximum implies a transition from positive wind shears below the peak to negative ones above. The effect that such transition inflicts on wind turbines has not been thoroughly studied. High-frequency data of actual atmospheric LLJs were used as input

to the NREL aeroelastic simulator FAST code, and simulations were performed with different vertical distances between the LLJ peak and the wind turbine hub. It was found that the presence of negative wind shears at the heights of the turbine appeared to exert a positive impact in reducing the motions of the nacelle and the tower in every direction, with oscillations reaching a minimum when negative shears covered completely the turbine sweeping area. Only the tower wobbling in the spanwise direction was amplified by the negative shears; however, this occurred at slower velocities and accelerations. The forces and

moments were also reduced by the negative shears. The aforementioned impacts were less beneficial in the rotating parts such as the blades and the shafts. Finally, the power output was slightly more stable. Those findings can be very important for the next generation of wind turbines as they reach deeper into the heights of more LLJs.

## 1   Introduction

Nocturnal Low Level Jets (LLJs) are defined as relative maxima in the vertical profile of the horizontal wind velocity. They

are produced by the stable stratification in the lower atmosphere and the inversion of potential temperature that are mainly detected at night. LLJs occur in many regions around the world and are often observed in the Great Plains of the United States. They are particularly important due to their role in the formation of the climate and their impacts on the production of wind energy. Wilczak et al. (2015) determined that LLJs drive wind farm' capacity factors to over 60% during the nocturnal hours. Thus, they are beneficial for the wind energy production; however, it is not totally clear what is their influence in terms of the

structure.

There is no single mechanism to explain why LLJs occur. Blackadar (1957) was the first who explained the formation of the jets as the result of the inertial oscillation of the Earth. The periodicity of the inertial oscillation was later calculated by Stensrud (1996) and Van de Wiel et al. (2010). Additionally, other theories of LLJ's formation have also been proposed.





Holton (1967) noticed that a sloping terrain may have an influence in the dynamic forcing of a jet. Bonner (1968) confirmed the finding but maintained that the inertial oscillation was still the predominant mechanism and refined the conceptual model of the oscillations. Another theory was later proposed, when Uccellini (1980) observed that strong jets in the upper layers of the atmosphere can induce slower jets within the atmospheric boundary layer.

The most distinctive feature of a LLJ is a peak in the vertical profile of the horizontal wind velocity, usually appearing between 100m and 700m above the ground level, as noticed by Stensrud (1996). Gutierrez et al. (2016) observed that LLJs exert a noticeable impact at altitudes as low as 40m which indeed results in a direct influence over the performance of wind turbines. The existence of the velocity peak implies that the wind speed shear, defined as the variation of the wind speed with the height above the ground level, is positive below the jet peak and negative above.

Due to the increase in wind speed, LLJs are significant contributors of wind energy. Gutierrez et al. (2016) noticed an increase in wind power density in the order of 10 - 15 times the values in diurnal unstable conditions. On the other hand, Kelley et al. (2004) and Kelley (2011) demonstrated that there is an increase in mechanical loads and fatigue loads in presence of LLJs.

     Gutierrez et al. (2016) pointed out that, as wind turbines get taller, they reach deeper into the atmospheric layers where LLJs
are observed, and this transition from positive wind shear below the jet peak to negative wind shear above will be found more frequently near, inside or even below the turbine's sweeping area. The effect that such transition inflicts on wind turbines has not been thoroughly studied. Therefore, the key objective of this article is to determine the impacts of the wind shear transition over the motions (deflections, velocities and accelerations) and loads (forces and moments) of the turbine's parts. Differently from rotating parts (i.e., the blades and the shafts), the set of nacelle and tower is anchored at one end to the ground, which
tends to increase the forces and bending moments at the points of support. This situation leads to deflections at the nacelle and at the top of the tower, and considerable forces and moments at the base of the tower. Therefore, it is of interest to find if the LLJs shear transition exacerbates or mitigates those effects.

     The rest of the paper is organized as follows. Section 2 focus on literature review about the current state of knowledge on wind data collection and on wind shear. Methods are described in Section 3, including how data were collected and processed,
how simulation cases were prepared and how turbine simulations were performed. Results are shown is Section 4 with the most characteristics outputs in the turbines' blades, nacelle and tower. Finally, in Section 5 we discuss the results and present major conclusions.

## 2    Literature review

As the size of wind turbines continue to grow, they reach deeper into the heights where LLJs are more frequently found, as
noticed by Gutierrez et al. (2016) Therefore, the confluence of positive shears below the peak of the jet and negative shears above will be found more often within the sweeping area of the next generation of wind turbines.



## 2.1 Data collection methods

To simulate the results of this interaction between turbines and LLJs, a decision must be made about how to model the LLJ.
Two trends are available: obtain direct measurements of real wind data or generate synthetic data from computational models.
The first option is usually implemented through costly measurement infrastructures, including meteorological towers, Sonic

Detection And Ranging (SODAR) devices, Light Detection And Ranging (LIDAR) devices or surface stations. Due to high
costs associated with field campaigns, the trend is to create synthetic data using economical computer models. The first task in
this research was to clarify the current accuracy of these technologies in order to select the best option to simulate LLJs.

The attempts to model atmospheric events, including LLJs, are not new. Storm et al. (2009) conducted a research to assess
whether Numerical Weather Prediction (NWP) models can substitute expensive equipment such as meteorological towers

in forecasting LLJs. They found that the Weather Research and Forecasting (WRF) model was able to capture some of the
main characteristics of the observed LLJ events; however, WRF proved to be inaccurate in predicting important LLJ features
such as peak height and speed. Further attempts were performed by Storm and Basu (2010) to evaluate whether WRF is a
significant improvement in the estimation of the shear coefficient, which in wind energy projects is estimated as $\alpha = 1/7$.
Results demonstrated that WRF can provide a better approximation; however, their applicability to accurate projects was

uncertain, as other factors such as terrain and canopy were not considered.

While modelling coastal LLJs, Nunalee and Basu (2013) observed that the model accuracy depended heavily on the selection
of parameters and initial conditions, and that the estimates of the annual energy production (AEP) varied widely. More recently,
Van de Wiel et al. (2015) noticed that WRF simulations tended to underestimate the jet intensity. It can thus be concluded that
NWP models based on WRF are promising at present state, however more progress is needed to consider these models as

reasonable options in detailed analysis of mechanical impacts on turbines.

Attempts have also been made at considering other factors in the models. For instance, Choi et al. (2011) investigated
whether the characteristics of the terrain influence the dimensions of the atmospheric boundary layer (ABL) significantly. This
is rather important in simulations of LLJs, as they occur in the upper part of the ABL. Their findings revealed that the height
of the ABL in a forested terrain, which is directly related to the integral length scale of the turbulent flow, could be estimated

using an interpolation formula. The parameters in their formulation confirmed that there were factors involved in the generation
of the wind shear that were not considered in usual simulations.

Other trends have been pursued in the numerical simulations of LLJs. Werth et al. (2011) compared results from a numerical
mesoscale model (RAMS) against real wind data from actual LLJs events. The spatial and temporal resolutions were too
coarse to simulate transient responses on wind turbines, but the experimental part provided statistics of some of the most

relevant features of LLJs. The same year, Sim et al. (2009) used locally-averaged scale-dependent dynamic (LASDD) model
to generate stable conditions' wind data in order to assess whether turbine loads could be calculated accurately. They obtained
fatigue loads in stable conditions that were 10% lower than those observed in neutral conditions. However, the uncertainty was
rather high.



Wilczak et al. (2015) worked on the Wind Forecast Improvement Project (WFIP) to evaluate whether wind energy can be forecasted in the medium term (18 - 42 hours) and in the short term (0 - 6 hours). Results showed that LLJs still represented a challenge for numerical modelling due to parametrization inaccuracies, although improvements are continuously made. On the other hand, Park et al. (2015) performed a large-eddy simulation (LES) study to assess the adaptability of traditional models

based on neutral boundary layers (NBL) to describe stable boundary layers (SBL). They found that such technique did not provide a unified model able to match the resulting turbine loads during different stability scenarios. More recently, Pichugina et al. (2016) confirmed that the power law was a bad predictor of wind speed in a LLJ, even in the positive shear zone below the jet peak.

In contrast to numerical methods, experimental data acquisition methods appear to be better in capturing LLJ information.

For example, Madougou et al. (2012) used UHF radars to evaluate whether a region from West Africa to Central Africa was suitable for wind energy purposes. They found that although winds were weak at the ground level, strong nocturnal jets were found to be providing enough energy to drive the wind turbines. The spatial and temporal resolutions of the UHF radars were appropriate for the purpose of their research; however, they lacked the frequency needed for more sensible studies such as the mechanical simulations included in the present article.

Instruments installed on meteorological towers usually provide data with less uncertainty. One advantage of using meteo-rological towers is the ability to concentrate measurements closer to the ground. Ferreres et al. (2013) investigated whether tower observations could capture the main features of several coherent structures found in atmospheric stable conditions, one of those structures being a Low-Level Jet (LLJ). Although the spatial and temporal resolutions of the instrumentation were not high, the structures were correctly detected and analyzed by using wavelength methods. The study demonstrated that the

towers were a reliable way to capture accurately the features of such events, including the LLJ.

The literature indicates that high-frequency instruments installed on meteorological towers are the best option to capture the scales of wind motions that structurally affect wind turbines. Therefore, the strategy in this research was to process high-frequency wind data obtained from a 200-m meteorological tower, then use them as input to an aero-elastic simulator, and learn about the turbine's response.

**2.1.1  Previous work on wind shear**

The implications of the LLJs' increased wind shear on the performance of wind turbines have been investigated in recent years. Antoniou and Pedersen (2009) studied the possible correlations between wind speed shear and annual energy production (AEP) for a wind turbine in Midwest, USA. They documented that the wind shear influenced the turbines' power curve strongly. They also detected stronger wind shears, lower turbulence intensity and increased AEP at night, which were attributed to the

nocturnal LLJ. Similar results were obtained by Antoniou et al. (2009) who extended the analysis to the wind farm by using measurements from the Danish Test Station for Large Wind Turbines, located in Denmark. They also detected a stronger influence of the LLJs over the upper section of the turbine rotor. Greene et al. (2009) correlated measurements of wind data in Western Oklahoma with the power production of three commercial wind turbines in the same area. The actual power outputs



in LLJ's situations were found to be larger than the traditional estimates due to stronger wind shears. Measured turbulence intensities were generally lower when LLJs were present.

A comprehensive study about LLJs- induced damage on wind turbines was conducted by Kelley et al. (2004). The research was based on the collection of real wind data from a tower and a SODAR, and also on measurements of resulting loads on
real wind turbines. They found that LLJs can cause instabilities leading to coherent turbulence and Kelvin-Helmholtz waves, which they correlated with an increase in flapwise loads in the blade roots. In a following study, Kelley (2011) detected that the maximum turbine damages occurred within a narrow range of atmospheric stability usually associated with LLJs.

Sathe et al. (2013) used wind profile models together with an aerolastic simulator to investigate whether the wind profile and the atmospheric stability modify the wind turbine loads. They found that loads at different turbine parts were affected slightly
by the wind profile (up to 7%) and significantly by the atmospheric stability (up to 17%). On the other hand, Bhaganagar and Debnath (2014) performed large-eddy-simulations (LES) to understand how the stable atmospheric boundary layer (ABL) affects the wind turbines' wake in a wind farm. They found that the wind speed shear, the wind direction shear and the atmospheric turbulence were the parameters with more influence over the structure and lateral expansion of wakes.

Walter (2007) demonstrated the importance of the wind speed shear and the wind direction shear for the wind turbine's power
production. Specifically, he showed a \$2.1 million loss in revenues across the lifetime of a 100 MW wind turbine, compared to revenues calculated for the baseline case with no shears. More recently, Hur et al. (2017) included the strong wind shear as one of the anomalies in the incoming wind field that an enhanced turbine controller needs to detect in order to compensate the mechanical imbalances on the turbine structure. Gutierrez et al. (2016) pointed out an exacerbation of the cyclical loads on the blades as a result of the stronger wind shear below the LLJ peak, which roughly increased the shear coefficient around five
times.

A main limitation of the aforementioned studies is that they were focused mostly on LLJ's strong positive wind shears as a whole, which basically means that shears were considered to be entirely positive across the turbine's rotor. To our knowledge, no research has been performed about the possible effects of the presence of LLJs' negative wind shears within the turbine's sweeping area. This scenario is becoming more important as the wind turbines get larger, reaching more often the heights of
LLJs' peak. Giammanco and Peterson (2005) studied LLJs in the region of the present research, and filtered examples whose velocity peak occurred below 200m above the ground level. They detected characteristic features of LLJs, including extended periods of low turbulence interrupted by bursts of turbulence that were associated with Kelvin Helmholtz instabilities. On the other hand, Zhou and Chow (2012) observed that strongly stable boundary layers (with strong surface cooling) tended to generate the jet peak at lower heights than those created by moderately stable boundary layers. This observation indicates that
strong jets are possible at low altitudes above the ground level.

In summary, the action of negative shears within the turbine's sweeping area is a phenomenon whose consequences have not been thoroughly studied. Understanding their effects on the performance and mechanical loads of wind turbines is of great importance from an operational point of view, as the knowledge can modify the expectations and assumptions included in the design of future wind turbines.





## 3   Methods

Measurements of wind speed, temperature, pressure and relative humidity were continuously collected by West Texas Mesonet (2017) station at Reese Technology Center. The data were processed and consolidated into a common database. To prepare the simulation cases, the database was screened for patterns indicating atmospheric conditions like LLJs. Once a segment of data was found to match the pattern of interest, it was prepared for a simulation. The dataset was provided as input to an aero-elastic simulator program that computed the mechanical responses of a wind turbine to the incoming wind field.

The first part of this section describes in detail the main features and configuration of the measuring devices mounted on the 200-meter's meteorological tower. The second part explains the processing of the data and how a case was selected. The third part explain how simulations were prepared for different jet altitudes with respect to the turbine height. Finally, the fourth part describes the features of the aero-elastic simulator and how the simulations were performed.

### 3.1   Data collection

The bulk of the experimental data was collected from the measurement system of the West Texas Mesonet 200- meters' meteorological tower described by Hirth and Schroeder (2014). The tower is located at N $33°36'27.32''$, W $102°02'45.50''$ and at elevation of 1021m. Sensors were installed at 10 vertical positions along the tower as follows: 0.91m, 2.44m, 3.96m, 10.06m, 16.76m, 47.24m, 74.68m, 116.43m, 158.19m and 199.95m. All tower measurements and dependent parameters were obtained at a frequency of 50 Hz.

Gill R3-50 sonic anemometers at each height were used to obtain the measurements of the three components of the instantaneous velocity: $u$ (northward), $v$ (eastward), and $w$ (vertical). The horizontal wind, which was later considered to be normal to the plane of rotation of the blades at the height of the turbine hub, was obtained as the vector sum: $\boldsymbol{U}_{xy} = \boldsymbol{u} + \boldsymbol{v}$. The modulus of the horizontal velocity was calculated as $U_{xy} = \sqrt{u^2 + v^2}$.

At each height, Young 41382VF sensors provided measurements of temperature $T$ and relative humidity $RH$, while Young 61302V barometers measured the atmospheric pressure $P$. The potential temperatures were then calculated as follows,

$$\theta = \left(\frac{P_0}{P}\right)^{R/c_p} \tag{1}$$

where $P_0$ is a pressure reference, $R$ is the gas constant of air, and $c_p$ is the specific heat capacity at constant pressure.

The virtual potential temperatures were calculated using,

$$\theta_v = \theta(1 + 0.61r) \tag{2}$$

for unsaturated air, where the mixing ratio $r$ was obtained from the relative humidity, as described by Stull (1988).

### 3.2   Data processing

The data obtained from the tower system were periodically processed and transferred into a central database optimized for complex queries. Data processing included the elimination of corrupted, incomplete or inaccurate records, the conversion to





a normalized data structure, the removal of superfluous information, and the calculation of dependent parameters. The central database was then scanned in search of past atmospheric events, including the occurrence of LLJs incidents. LLJs cases were detected by using a trigger combining: high vertical shear of horizontal wind speed, low standard deviation of wind speed at the peak height, and sustained winds for at least 12 hours. Preference was then given to LLJs cases whose peak in the horizontal

velocity profile occurred below 200 m above the ground level.

After applying these filters, it was determined that the best example of a strong LLJ at low height was the incident that occurred on 22-23 October 2013. Finally, 30-minutes samples from the selected case were spatially interpolated to obtain 3D series with time resolution of 0.02 s (50 Hz) and space resolution of 1 m in both directions.

### 3.3  Case preparation

The main goal of this article is to compare the impacts over the turbine's parts inflicted by the negative wind speed shear above the LLJ peak, in relation to the impacts caused by the positive shears below the peak. A dimensionless parameter was used to quantify the proportion of the turbine rotor that received negative wind shears. The turbine-jet relative distance parameter was defined as follows,

$$\xi = \frac{(z_t - z_p)}{R} \tag{3}$$

where $z_t$ is the height above the ground level of the turbine hub, $z_p$ is the height above the ground level of the peak of the jet, and $R$ is the turbine rotor radius. Characteristics values of the parameter $\xi$ are represented in Figure 1.

Values of the parameter $\xi$ were generated by two methods, as shown in Figure 2. In the first method, values of $\xi$ were created by simulating the turbine at different heights, and plugging in the wind data grid into each simulated turbine. This procedure was usually preferred as it was computationally less demanding; however, a correction was needed to compensate for the

variation of bending moments at the tower attributable to the increase of tower height. In the second method, values of $\xi$ were obtained by creating a family of synthetic data grids derived from the original jet data. Each member of the family was made to peak at different heights while keeping constant the total kinetic energy of the wind in the region of the turbine's sweeping area. In this method, the turbine height was kept constant for all simulations.

### 3.4  Impact simulation

The structural responses of the wind turbine are defined here as the three-dimensional outputs including: deflections, velocities, accelerations, forces, torques, and other results occurring at several parts of the turbine as a consequence of the action of an incoming wind. The wind turbine components analyzed were the blades, the low-speed shaft, the high-speed shaft, the nacelle and the tower. The structural responses were obtained by plugging in the data grids into the FAST (Fatigue, Aerodynamics, Structures, and Turbulence) simulation code developed by NREL National Wind Technology Center (2016). FAST

is a comprehensive aero elastic simulator capable of predicting both the extreme and fatigue loads of two- and three-bladed horizontal-axis wind turbines (HAWTs). Simulations in this article were performed with the NREL WindPACT 1.5-MW Wind Turbine described by Malcolm and Hansen (2006).





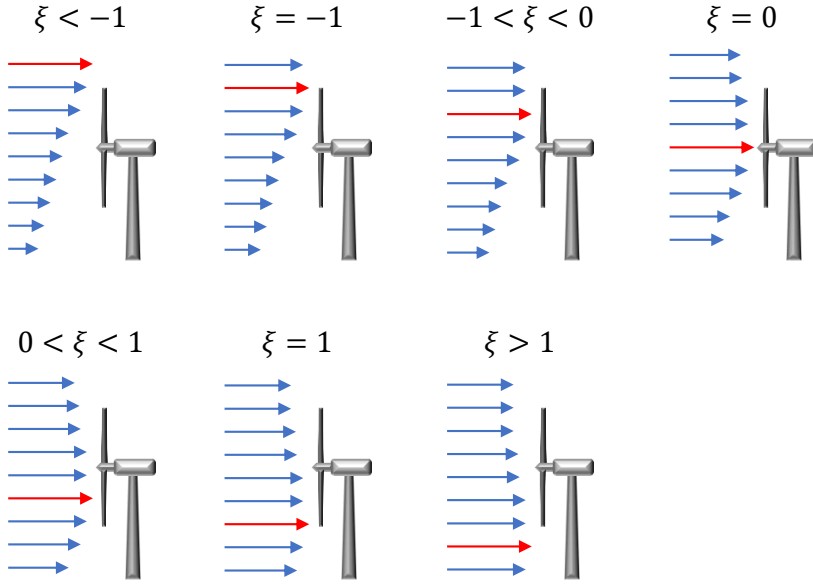

**Figure 1.** Characteristic values of the turbine-jet relative distance $\xi$. The LLJ, represented with blue arrows, impacts the turbine at different vertical positions. The red arrows depict the peak of the jet.

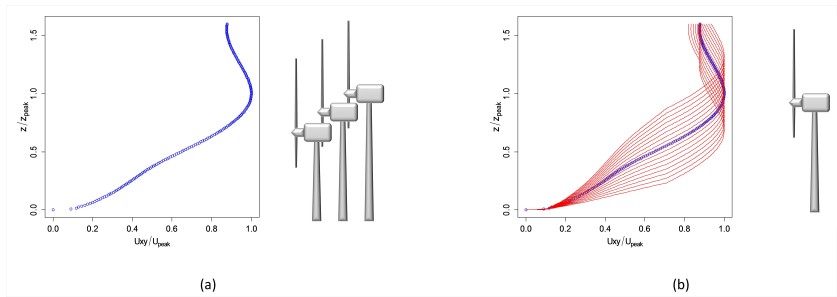

**Figure 2.** Methods to generate values of the parameter $\xi$. In the first method (a) a single wind data grid was combined with the turbine simulated at several heights. In the second method (b) the turbine at a single height was combined with a family of synthetic data grids (represented in red lines) generated after the original jet data (in blue line).

## 4   Results

The results from each FAST run were composed of time series of deflections, velocities, accelerations, forces, moments and other outputs in different parts of the wind turbine. In this article, each of those time series is called a turbine response. A total




of 379 responses were obtained and analyzed for each value of the parameter $\xi$. The way the mechanical responses varied with the parameter can be analyzed by looking at plots of probability density function (PDF) for each response setting.

The rest of this section is organized as follows. In the first subsection, a detailed explanation is given to demonstrate how a 3D representation of the PDFs for every $\xi$ constitutes the best option to analyze the influence of the parameter on the turbine's
responses. Then, the remaining subsections analyzes the results obtained for a selected subset of turbines responses. Due to space restrictions, this article only shows those responses whose behaviors were the most interesting and representative of the overall conclusions.

### 4.1 3D representation of the PDFs

Each response value $R$ obtained after all FAST simulations corresponds to: a specific location in the wind turbine, a specific
type of response (such as a deflection, a force, etc.), the simulation at a specific value of $\xi$, and a specific time after the start of the simulation. The collection of all response values at a given location, of a given type and resulting from a given $\xi$ simulation is a time series called a response $r$. The statistical analysis on each time series allows to assess the average effect on the response of the incoming wind as the turbine rotor spins repeatedly across the wind field.

The PDF for any turbine response $r$ at a given value of $\xi$ is defined as the function $f_{(r,\xi)}$ that satisfies the following expres-
sion,

$$Pr[a \leq T \leq b] = \int_{a}^{b} f_{(r,\xi)} dr, \; \xi = const \tag{4}$$

where $Pr[a \leq T \leq b]$ is the probability of a response value $T$ of being within the interval defined by $a$ and $b$.

Figure 3a shows an example of PDF for a turbine's response, which in the figure is the streamwise tip deflection of the first blade. The x-axis (horizontal) represents the turbine response $(TipD_{xc}^{(1)})$ normalized by a reference $(D_0)$, while the z-axis
(vertical) shows the PDF. The analysis of the patterns observed for this response is performed in the next subsection.

As mentioned previously, the turbine's responses have been normalized in order to display dimensionless parameters in the x-axis. The normalization parameters were calculated in such a way that they were functions of turbine properties only. Their expressions are shown in the second column of Table 1.

The 2D plot of PDF for a single value of $\xi$ (Figure 3a) can be extended to a 3D plot that encompass all PDFs corresponding
to all values of the parameter, as seen in Figure 3b. The figure shows a second horizontal axis which represents $\xi$. This is greatly advantageous, because it allows to discover easily how the statistical response was affected by the parameter.

Nevertheless, a final transformation is highly recommended, as pure 3D plots may hide some useful information from the eyesight. The PDFs are best visualized by using density plots, as seen in Figure 3c, which shows the projection of the plot on the plane. In this final form, the values of PDFs are represented by the color depth, with darker colors indicating peaks of
PDF and therefore more concentration of values. An additional advantage is that now we can draw lines to connect important information across values of $\xi$, including means, medians, confidence intervals, etc.



**Table 1.** Normalization parameters. Values of turbine properties were obtained from Malcolm and Hansen (2006)

| Dimension type | Normalization parameter | Turbine properties |
|---|---|---|
| Translational deflections | $D_0^{(tra)} = R$ | $R$: rotor radius[a] |
| Shafts' rotations | $\Omega_0 = \Omega_{max}$ | $\Omega_{max}$: maximum angular speed[a] |
| Angular velocities | $V_0^{(rot)} = 6\Omega_0$ | |
| Translational velocities | $V_0^{(tra)} = V_n$ | $V_n$: nominal tip speed[a] |
| Angular deflections | $D_0^{(rot)} = \frac{360\sigma}{B}$ | $\sigma$: solidity ratio[a] $B$: number of blades[a] |
| Power | $Pw_0 = Pw_n$ | $Pw_n$: nominal power[a] |
| Pressure | $P_0 = \frac{\rho V_w^2}{2}$ | $V_w = V_n/\lambda$: nominal wind speed |
| | | $\rho$: air density[a] $\lambda$: tip speed ratio[a] |
| Forces | $F_0 = P_0 A$ | $A = \pi R^2$: turbine sweeping area |
| Moments | $M_0 = F_0 z_h$ | $z_h$: nominal hub height[a] |
| Translational accelerations | $A_0^{(tra)} = \frac{F_0}{M}$ | $M = M_r + M_n + M_t$ |
| | | $M_r$: rotor mass[a] $M_n$: nacelle mass[a] $M_t$: tower mass[a] |
| Angular accelerations | $A_0^{(rot)} = \frac{M_0}{I}$ | $I = (M_r + M_n + \frac{1}{4}M_t)z_h^2$: inertia |

[a]obtained from Malcolm and Hansen (2006)

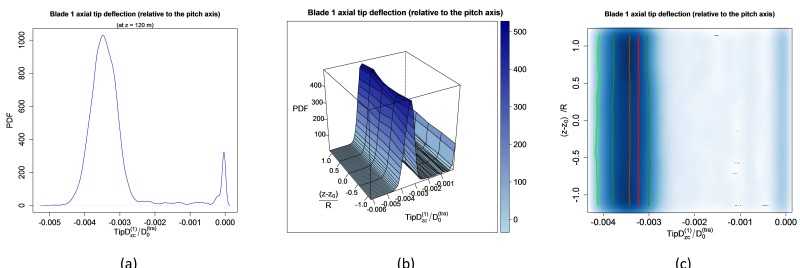

(a)  (b)  (c)

**Figure 3.** Generation of a density plot of PDFs. The streamwise tip deflection is used as example of turbine's response. (a) PDF at a single $\xi$. (b) 3D plot of PDFs for all $\xi$. (c) Color plot of PDFs for all $\xi$. The x-axis is the normalized response; the y-axis is $\xi$; the z-axis is the PDF.

The figures in the following sections (from Figure 6 to Figure 12) illustrate the PDF variation with $\xi$ of several responses. In each figure, the parameter $\xi$ is represented in the vertical axes, starting with shear totally positive at the bottom (where $\xi = -1$) and ending with shear totally negative at the top (where $\xi = 1$). The background color is deeper where values are more concentrated. The red line and the dark red line connect the mean and the median values respectively. The green lines and the dark green lines delimit the zones encompassing 95% and 68% of the values respectively.





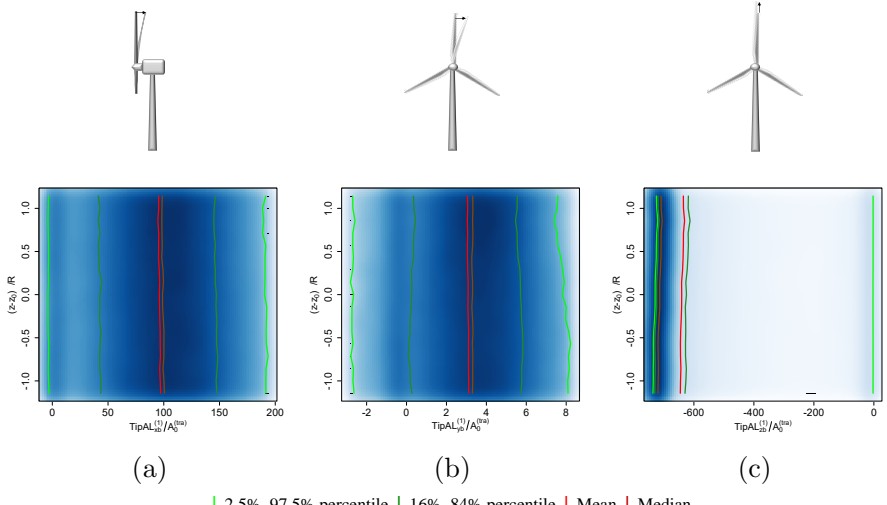

**Figure 4.** Variation with $\xi$ of the PDFs of the blade tip's translational accelerations. (a) Streamwise. (b) Spanwise. (c) Radial. Normalized deflections are in the horizontal axis, $\xi$ is shown in the vertical axis, while the PDF is represented as color depth (for details about the plot elements, see first subsection of the Results section.)

### 4.2  Blade's motions

The turbine's blades constitute a critical mechanical part of the whole system. Ribrant and Bertling (2007) published statistics of failures collected in Sweden from 1997 to 2005. They determined that the parts with most failures were the electric system, the sensors and the blades/pitch, in that order. On the other hand, they also observed that most downtime was attributed to the gearbox, the control system and the electric system, while the blades/pitch ranked fifth in a list of 13 turbine components. Overall, the blades/pitch component accounted for 13.4% of the failures and 9.4% of the downtime. The importance of the blades' damage is further boosted by their cost, as noticed by Li et al. (2015).

The increased wind speed and shear from LLJs augment the motions and loads on the blades, compared to the resulting motions and loads in a diurnal unstable atmosphere. Surprisingly, the nature of the LLJ' wind shear, i.e., the proportion of positive and negative shears, does not seem to exert a substantial influence on those motions and loads. This observation can be drawn from the analysis of Figure 4, which shows the PDF variation corresponding to the translational accelerations of the tip of one of the blades. The figure shows patterns of distributions that are almost symmetric for the streamwise and the spanwise components, and are very skewed for the radial component. As observed, the mean values of the three components remained almost constant and the variances decreased very slightly (especially in the spanwise component) when the parameter $\xi$ augmented.

The analysis of other responses corroborates the previous conclusion. The examination of the blade tip deflections (not shown) reveals that most of the time, the blade was bended near maximum values in the direction of the wind, with transient returns to zero deflection. Within the plane of rotation, the blade was bended 85% of the time opposite to the rotation, which





may be a consequence of the inertia. Finally, the radial deflection oscillated around a reduced value of length, with transient returns to zero. Increases in the proportion of negative shears within the sweeping area did not change the streamwise component and only reduced marginally the spanwise and the axial components.

### 4.3 Blade's loads

The analysis of forces and moments at the root of one of the blades (not shown) confirmed the previous conclusion (see section above) that the negative shears within the rotor had a rather marginal impact on the blades. They caused less extreme values of the streamwise shear forces and of the bending moments within the plane of rotation. They also reduced slightly the spanwise shear forces and the streamwise bending moments. Finally, more negative shears resulted in small reductions of the centrifugal forces and the torsional moments. The analysis of loads also revealed that the centrifugal forces were predominant, with values

around twelve times the values of the streamwise shear forces, and close to forty times the values of the spanwise shear forces.

### 4.4 Nacelle's motions

The motions of the nacelle were small in all three directions; however, they can be connected to the forces and moments that occurred at the base of the tower (see section below).

Figure 5 shows the variations of the PDF corresponding to the three components of the nacelle's translational accelerations.
It is observed that all components oscillated around the equilibrium position at zero values. The streamwise component resulted the most important, with values in the order of six times the values of the spanwise component and almost ten times the values in the vertical direction.

On the other hand, Figure 5 also provides insight into the effects of the negative shear over the nacelle translational accelerations. It is noticed that the amplitude of oscillations of each acceleration component decreased approximately one third when
the wind shear across the rotor area went from totally positive to totally negative. This pattern indicates that the presence of negative wind shears tends to soften the fatigue impacts inflicted on the nacelle, as they reduce the variance of motions and loads.

Similar conclusions can be obtained from the analysis of the nacelle translational velocities (not shown). As with the accelerations, the magnitudes of all three velocity components were very small and oscillated around zero values. The amplitude of
oscillations of the velocity components varied very little in the interval $-1 \leq \xi < 0$, and then decreased to minimum values at $\xi = 1$ (shear totally negative). Oscillations reached a maximum when the peak of the jet impacted directly at the height of the turbine hub, and a minimum when negative shears covered completely the turbine sweeping area.

While the nacelle's translational motions induce shear forces at the base of the tower, the rotational motions can be associated to the tower base's moments. Figure 6 shows the variations of the PDFs corresponding to the three components of the nacelle's
angular accelerations. The figure reveals that the component around the cross axis (roll) was the most affected, with values in the order of four times the acceleration values around the rotor axis, and forty times the values around the vertical axis. This pattern can be explained by the thrust inflicted in the direction of the wind.



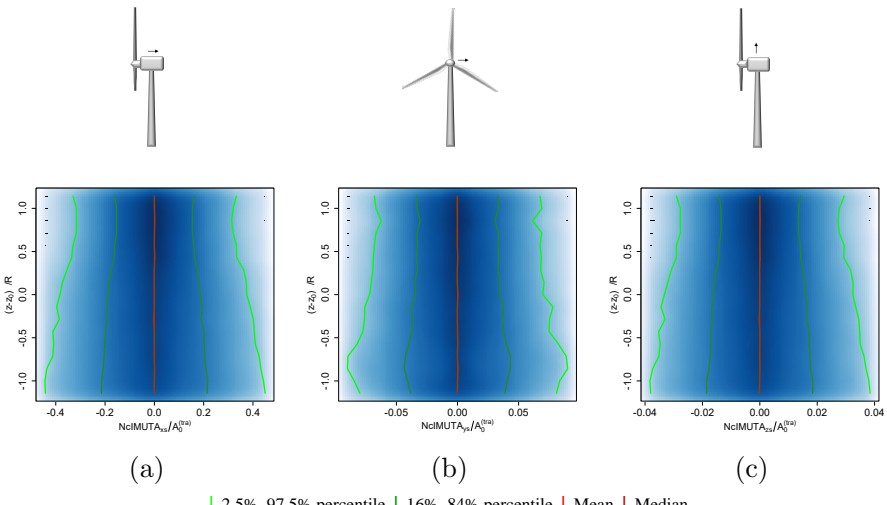

| 2.5%, 97.5% percentile | 16%, 84% percentile | Mean | Median

**Figure 5.** Variation with $\xi$ of the PDFs of the nacelle's translational accelerations. (a) Streamwise. (b) Spanwise. (c) Vertical. Normalized deflections are in the horizontal axis, $\xi$ is shown in the vertical axis, while the PDF is represented as color depth (for details about the plot elements, see first subsection of the Results section.)

Figure 6 also reveals a sharp decrease with $\xi$ of the amplitude of oscillations for the three components of the nacelle's angular accelerations. In fact, when the turbine rotor operated entirely in negative shears ($\xi = 1$), oscillations were reduced to only one sixth of the amplitudes observed when the turbine rotor was operated entirely in positive shears ($\xi = -1$). It can therefore be concluded that negative wind shears inside the turbine sweeping area had a strong dampening effect on the rotational motions

of the nacelle. In summary, the presence of negative wind shears at the heights of the turbine rotor appeared to exert a positive impact in reducing the motions of the nacelle in every direction.

### 4.5 Tower's motions

Due to its long and slim geometry, the turbine's tower is susceptible to considerable motions at the top, and large forces and moments at the base. The PDF plots of the tower top's deflections are shown in Figure 7. The distributions of the streamwise

component and the vertical component were concentrated close to their mean values, with transient returns to near-zero values. On the other hand, the spanwise component showed a back-and-forth motion between positive and negative values. Streamwise deflections were two orders of magnitudes greater than those in the other two directions. The magnitudes of those deflections were small; however, their fluctuating nature can contribute to the accumulation of fatigue cycles on the tower. In addition, the motions at the top can induce forces and moments at the tower base.

The plots in Figure 7 also show increases in the mean values of the streamwise component and the vertical component with the parameter $\xi$; however, those increases can be attributed to the modelling (enlargement of the tower) rather than to larger area under the influence of negative shears. On the other hand, negative shears can rightfully be attributed for the amplification of the oscillations that is observed in the spanwise component.





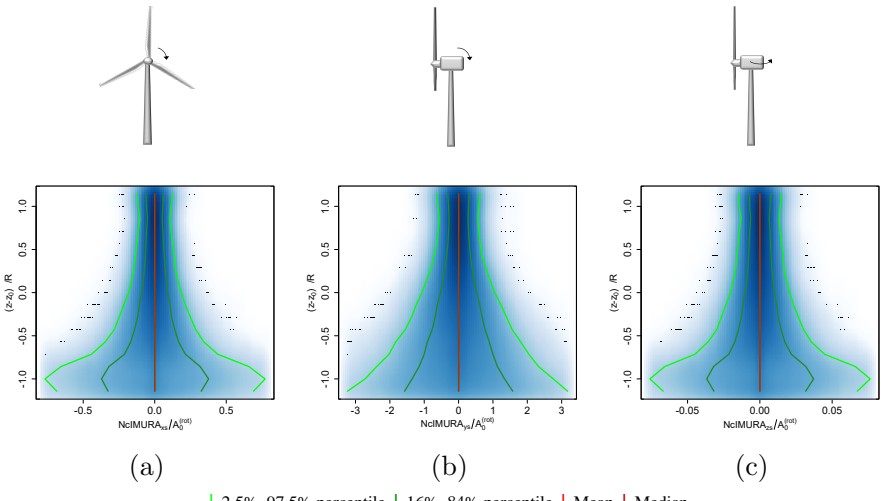

**Figure 6.** Variation with $\xi$ of the PDFs of the nacelle's angular accelerations. (a) Around rotor axis. (b) Around cross axis. (c) Around vertical axis. Normalized deflections are in the horizontal axis, $\xi$ is shown in the vertical axis, while the PDF is represented as color depth (for details about the plot elements, see first subsection of the Results section.)

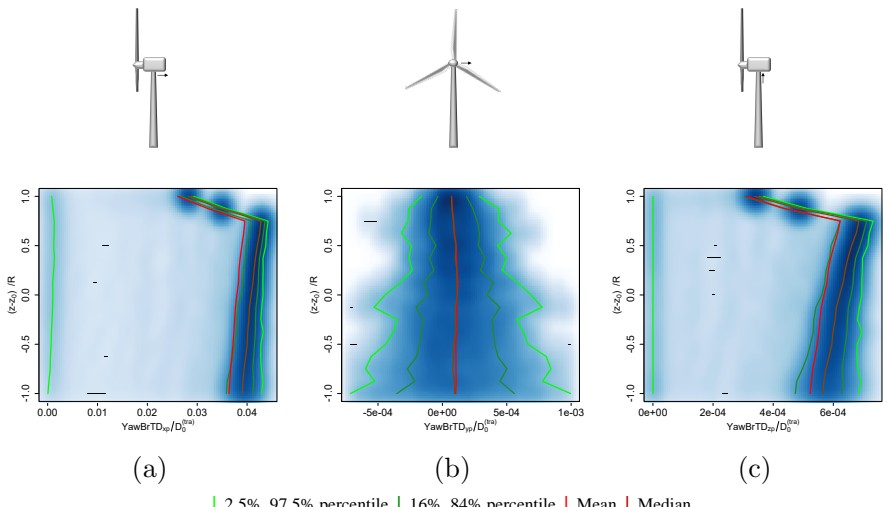

**Figure 7.** Variation with $\xi$ of the PDFs of the tower top's deflections. (a) Streamwise. (b) Spanwise. (c) Vertical. Normalized deflections are in the horizontal axis, $\xi$ is shown in the vertical axis, while the PDF is represented as color depth (for details about the plot elements, see first subsection of the Results section.)

The previous observations pointed out an increase in the amplitude of oscillations of one deflection component. Nevertheless, the potential damage was probably overcompensated by decreases in the oscillation frequencies, as revealed by Figure 8 which shows the PDF variations of the tower top angular velocities. It is observed that the angular velocities actually decreased




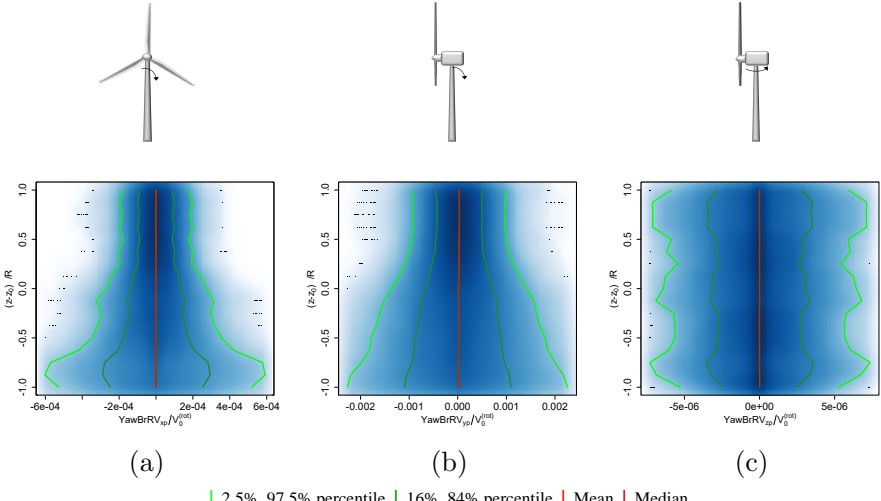

**Figure 8.** Variation with $\xi$ of the PDFs of the tower top's angular velocities. (a) In the plane of the rotor. (b) Across the plane of the rotor. (c) Torsional. Normalized deflections are in the horizontal axis, $\xi$ is shown in the vertical axis, while the PDF is represented as color depth (for details about the plot elements, see first subsection of the Results section.)

when $\xi$ augmented, both in the plane of rotation and across the plane of rotation. In fact, the oscillation amplitudes of both components decreased to levels smaller than one third the values observed when positive shears covered the turbine sweeping area. As a result, the presence of negative wind shears at the heights of the turbine rotor helped to reduce the accumulation of fatigue cycles in the tower. However, the net effect of increased amplitude and decreased frequency is not clear. A future
fatigue analysis on components near this zone may clarify the overall effect of the antagonistic amplitude and frequency.

### 4.6 Tower's forces and moments

The critical part of the tower is located around the base, where the shear forces and bending moments reach maximum values. The PDF plots of the tower base's forces are shown in Figure 9. It is observed that the streamwise shear component was two orders of magnitude stronger than the spanwise component. The compression force was even stronger, by an additional order
of magnitude. The distribution of the streamwise component was concentrated around 300 N in the direction of the wind, with transient returns to near-zero values. On the other hand, the spanwise component fluctuated between the opposite sides of the tower. Finally, compression forces were concentrated very close to the mean values.

   Figure 9 also demonstrates that the increase of the area experiencing negative wind shears reduced considerably the amplitude of oscillations of both shear force components. The figure also shows that the mean values of the compression force
augmented with increases of the parameter $\xi$; however, this can be attributed to the modelling (enlargement of the tower) rather than to a consequence of more negative shears within the rotor area.



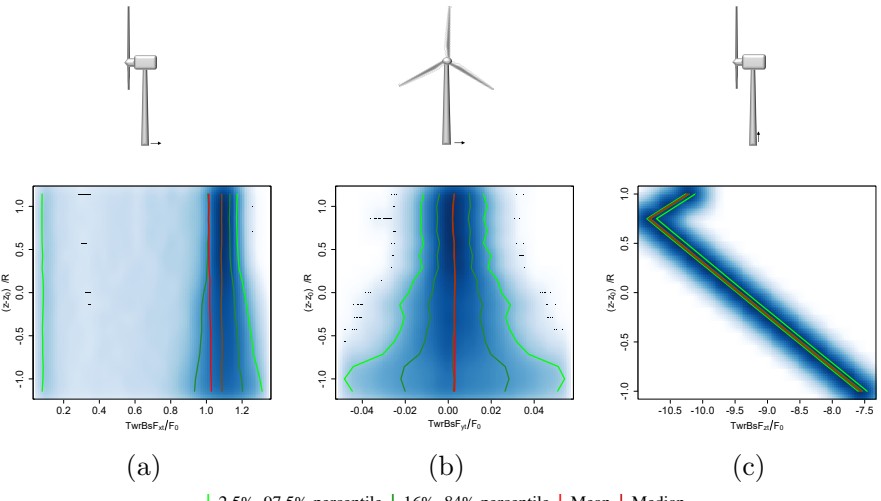

(a)  (b)  (c)

| 2.5%, 97.5% percentile | 16%, 84% percentile | Mean | Median

**Figure 9.** Variation with $\xi$ of the PDFs of the tower base's forces. (a) Streamwise shear. (b) Spanwise shear. (c) Traction-compression. Normalized deflections are in the horizontal axis, $\xi$ is shown in the vertical axis, while the PDF is represented as color depth (for details about the plot elements, see first subsection of the Results section.)

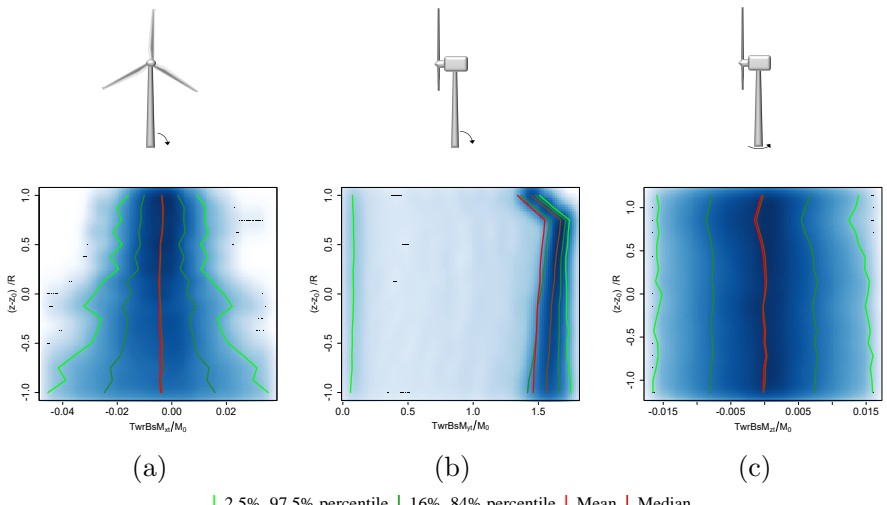

(a)  (b)  (c)

| 2.5%, 97.5% percentile | 16%, 84% percentile | Mean | Median

**Figure 10.** Variation with $\xi$ of the PDFs of the tower base's moments. (a) Bending in the plane of rotation. (b) Bending across the plane of rotation. (c) Torsion. Normalized deflections are in the horizontal axis, $\xi$ is shown in the vertical axis, while the PDF is represented as color depth (for details about the plot elements, see first subsection of the Results section.)

The analysis of the PDF variation of the tower base's moments, shown in Figure 10, reveals that the values of bending moment across the plane of rotation were two orders of magnitude stronger than the values of the bending moment within the plane of rotation and the values of the torsional moment.



In addition, the amplitudes of oscillations of the bending moments in the plane of rotation and the torsional moments appear to be damped by the negative shears. The figure also shows increases in the mean values of the bending moment across the plane of rotation when the parameter $\xi$ augmented; however, they can be attributed to the modelling (enlargement of the tower) rather than to a consequence of more presence of negative shears.

It can be concluded that the presence of negative wind shears at the heights of the turbine rotor helped to reduce the amplitude of the oscillations of several forces and moments at the tower base.

## 5 Discussion and Conclusions

This paper investigates the mechanical impacts that the presence of LLJ's negative wind shears can have over several components of commercial-size wind turbines. As documented, the wind shear is a characteristic feature of LLJs that plays a key role in the mechanical responses of wind turbine's parts.

High-frequency measurement instruments mounted on the meteorological tower provided wind data with sufficient accuracy to capture the time evolution of the wind speed impacting the wind turbine. It was assumed that the turbine control systems acted fast enough to keep the turbine sweeping area always perpendicular to the main wind direction. The assumption is valid since the direction of the wind in the presence of LLJs varied very slowly over time.

A non-dimensional parameter $\xi$ was created to quantify the proportion of negative wind speed shears reaching the wind turbine's rotor. Mechanical responses were determined for different $\xi$ values. Results showed that the transition from positive to negative shears had a weak-to-moderate influence over the amplitude of oscillations of several of those responses.

First, the proportion of negative wind shears within the turbine sweeping area had a limited impact on the motions and loads at the blades, with very small reductions in the variances of several deflections, accelerations, forces and moments. Although not shown, similar patterns were detected in the responses of other rotating parts such as the low-speed shaft and the high-speed shaft.

Second, the negative wind shears had a positive impact in reducing the motions of the nacelle in every direction. The effect was proportional to the ratio of the turbine's rotor that received negative shears. Variances reached minima when negative shears covered the turbine sweeping area completely.

Third, it was observed that even as the amplitude of deflections at the tower top increased slightly in the spanwise direction, the tower motions were slower when $\xi$ augmented. This indicates that the presence of negative shears may help in reducing the accumulation of fatigue cycles. However, the net effect of increased amplitude and decreased frequency is not clear. Future wear analysis on components near this zone under different $\xi$ values can clarify the overall effect of the antagonistic amplitude and frequency.

Finally, it was detected that the presence of negative wind shears at the heights of the turbine rotor reduced the amplitude of oscillations of several forces and moments at the tower base.



In summary, the negative wind shears, when present within the turbine's sweeping area, improved the mechanical loading of the turbine's nacelle and tower, as those shears were connected to a tendency to alleviate the amplitude and frequency of several motions and loads.

There are two factors that may explain the beneficial effects of the presence of LLJ's negative shears at the heights of the turbine rotor. First, the absolute values of slope of the wind profile above the peak of the jet are generally lower than the absolute values of slope below the peak. Therefore, the magnitude of the torque created by the negative shear becomes smaller than the one created by the positive shear. As a result, the negative shear above the peak generates smaller forces and moments.

The second factor is the difference in the distributed loads (i.e., the forces and moments generated on the wind turbine structure per unit height). When the wind shear is positive, the distributed loads (and therefore the localized impacts) augment with height. The worst scenario occurs when positive shears cover entirely the turbine. On the other hand, negative wind shears decrease the magnitudes of the distributed load. Because the wind turbine is fixed to the ground, the loads generated in the negative wind shear section are still added to the total loads; however, they are added at a smaller pace. As a result, the presence of negative wind shears at the heights of the turbine rotor actually relieves the operation loading of the wind turbine, compared to scenarios with no negative wind shears.

One seemingly surprising result was that the presence of negative wind shears was less influential on the rotating parts (such as the blades and the shafts) than on the static parts (such as the nacelle and the tower). Apparently, the rotating components of the turbine compensate the effects of different wind shears as they cycle through the wind field. In that case, the mechanical response of those rotating parts may be more responsive to the magnitude of the wind speed rather than to the values of wind shear. This contrasts with the responses of the static parts that this research observed to be more connected to the magnitudes of the wind speed shears.

The mean value of power production (measured in the low-speed shaft) remained basically constant with variations of the parameter $\xi$. On the other hand, its variance was slightly reduced with more presence of negative shears. Both observations highlight additional advantages. First, the presence of the negative shear maintains intact the enormous potential for power production of the LLJ, estimated by Gutierrez et al. (2016) to be in the order of 10-15 times the values in regular unstable conditions. Second, the negative shear sustains that power while also reducing slightly the probability of damaging loads. Finally, the reduction in variance may decrease the amount of power transients that the turbine's controller need to handle.

Thus, based on the results from this study, building taller wind turbines in which their rotors are within the negative shear region of LLJ's will benefit from high energy production with minimum external loads from wind shears on the wind turbines.

*Code availability.* Simulations in this research were performed using the aerolastic simulator FAST code, which is available to the public at the website of the NREL National Wind Technology Center (2016)

.



*Acknowledgements.* The authors gratefully acknowledge the following grants for this research: NSF-CBET #1157246, NSF-CMMI #1100948, and NSF-OISE-1243482.




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
