# Peer review of "Impacts of the Low-Level Jet's Negative Wind Shear on the Wind Turbine"

_Wind Energy Science, 2017_

## Referee Comment (RC1) · Anonymous Referee #1 · 18 Jul 2017

Dear authors,

Thanks for your manuscript. In general I think that it contains good and interesting information but in its current version it reads more like a technical report rather than a scientific contribution. I hope you answer all my major and specific comments, which hopefully help improving your work to the standard of a scientific paper.

Major comments

1. The literature review is way too long. Also you have a whole section with two subsections dedicated to it besides the introduction where there is also some literature review. My suggestion is that the literature review should be part of the introduction and you need to refer to the important work on the matter (you have many references

to non-valuable work). But please but do not make it much longer than it is!!!

2. You use the met tower data as an input for your aeroelastic computations. But from a tower (with different sensors at different vertical levels) you can derive a "2D" wind field. What do you assume to get the 3D wind field or how do you get it anyway?

3. As I mentioned before, I have the opinion that this manuscript reads more like a technical report than a paper for a scientific journal (the only exception is the discussion part that tries to give value to the work). The authors do not state clear how innovative their approach is. It just seems quite standard to perform aerolastic simulations using wind fields with certain characteristics.

4. As mentioned in the discussion the negative shears can have a smaller impact on the loads than the positive shears because the positive shears are larger in magnitude. I think it is important that after describing the met data, pdfs of the wind shear should be shown to see how different these are. Also, what will be the result if the magnitude of negative wind shears was as big as the positive ones? That will be interesting to see in the simulations

5. During LLJ conditions the variation of the wind shear might be large but so the wind veer and that might have a large(r) impact on the loads. This needs to be addressed in the manuscript.

6. Finally the authors repeat too many sentences throughout the manuscript than can be deleted without losing connection or detriment. I will point to some in the specific comments

Specific comments

1. Page 1 line 6 "at the heights of the turbine" For the abstract to stand alone, you need to say at what heights relative to the turbine hub height you refer to here.

2. Page 1 line 9 add "the tower's" before "slower velocities" otherwise the reader might think you are talking about the flow

3. Page 1 line 11 "slightly more stable" is quite ambiguous. Perhaps the whole sentence is not needed

4. Page 1 line 12 replace "the heights of more LLJs" by "LLJ's typical heights"

5. Page 1 lines 15-16 replace "are mainly detected" by "often occur"

6. Page 1 line 19 shouldn't you move the second "is" after "influence"?

7. Page 1 line 20 add "turbine" before "structure"

8. Page 2 line 6 leave a small space between numbers and units throughout the whole manuscript (this I think it is the first instance)

9. Page 2 lines 28-end is not needed + already said. Remember to cleverly, efficiently and shortly combine sections 1 and 2!

10. Page 3 lidar and sodar can be used already as words so you can save the explanation of the acronyms

11. Page 3 line 12 "which in wind energy... 1/7" This is not true. If somebody is doing this they are doing it wrong. 1/7 is the shear for a specific roughness, height, and stability conditions (in flat terrain)

12. Page 3 lines 24-25 "could be estimated using an interpolation formula" If this is indeed true then nobody should refer to that paper at all because there are no physics in that

13. Page 3 line 30 add "a" after "used"

14. Page 4 "More recently... jet peak" The power law is not physical so the reference to that study is not needed at all. The strange thing will be that they find the power law to be applicable to LLJs

15. Page 4 line 9 the sentence is not needed as it is obvious. The strange thing will be the opposite

16. Page 4 line 19 you mean "wavelet" instead of "wavelength"?

17. Page 4 lines 21-22 this sort of conclusion is not true. You refer to some literature regarding measurements with remote sensors that are not valuable for comparison. There are many papers where winds (and LLJs) have been observed with lidars (for example) and that you are not referring to

18. Page 4 line 29 "AEP at night" I guess you mean production... AEP is in a yearly basis and is one single number

19. Page 4 line 30 "to the wind farm" which wind farm? You are referring to some work from Antoniou which is not that much relevant. The danes have made many other campaigns with lidar measuring the wind profile all the way up to the ABL height that are relevant for your study

20. Page 5 line 15 you mean "farm" instead of "turbine"?

21. Page 6 line 9 replace "explain" by "explains"

22. Page 6 line 8 and 12 replace "200-meter's" by "200-m"

23. Page 6 line 27 where in Stull (1988)?

24. Page 6 lines 29-30 "The data... queries" is not needed. Also how you know they are "corrupted" or "inaccurate"? "the conversion to a normalized data structure... paremeters" is not needed either

25. Page 7 line 4 what do you mean by "sustained winds" and by "preference" (you use them or not)?

26. Page 7 line 8 "resolution of 1 m in both directions" which 2? It is just tower data! If you meant the turbulence box then you need to say how you go from a tower to a turbulence box where you need to model the three wind speed components. What assumptions you made for coherence?

27. Page 7 lines 10 -11 Not needed, already said

28. Page 7 line 15 replace "height above the ground level of the turbine height" by "turbine hub height"

29. Page 7 lines 25-27 Not needed, already said

30. Fig. 2 The axes in your figures cannot be read. You need to increase the size of the labels of nearly all figures

31. Page 8 After "Results" this is repeated all over the manuscript. Delete!

32. Page 9 line 2 replace "by looking at plots of" by "with"

33. Page 9 lines 24-31 this information is not needed, is too elemental about pdfs and plotting. Also in figure 3 frames b and c contain the same information. Cannot read the axes labels!

34. Page 10 all the text is not needed and/or has been said before. The description of what the lines are should be in the caption

35. Page 11 lines 2-7 This can be omitted

36. Page 11 lines 8-9 Is this a statement or an observation from your analysis?

37. Page 11 line 13 "are very skewed" where? I cannot see skewness

38. Page 11 line 16 change "conclusion" by "observation"

39. Page 12 line 5. Similar to comment 38

40. Page 12 lines 12-13 Not needed, delete please

41. Page 12 lines 18-19 similar to comment 40

42. Page 12 line 23 similar to comment 38

---

## Referee Comment (RC2) · Anonymous Referee #2 · 27 Jul 2017

Referee Comment on WES-2017-22 "Impacts of the Low-Level Jet's Negative Wind Shear on the Wind Turbine" by Gutierrez et al.

The manuscript investigates the impact of so-called low-level jet (LLJ) atmospheric flow situations on the loads induced on horizontal axis wind turbines (WT). Appropriate measurement data periods of a 200 m met tower are selected and used to generate input wind fields for FAST simulations. Resulting loads from these simulations are compared with respect to the vertical position of the LLJ relative to the WT hub height. As a main result, loads on non-rotating components are generally reduced when negative shear is induced by a LLJ within the rotor area.

The paper addresses a topic of high interest, especially for possibly larger WTs of the future. The data and methods used are - as far as explained - sound and consistent,

and the results are relevant. Publication of the manuscript in a revised form is therefore recommended.

General comments

1. Alltogether, the manuscript is lengthy. The authors should consider to identify the key cases and results and concentrate on these. Also the literature review, while much appreciated, appears to be more extensive than necessary.

2. The manuscript reads quite technical and should be improved in terms of a dedicated research question and a rigorous sequence of steps and arguments to give answers to that question. Most of the necessary elements are already mentioned in the paper, but they should be better worked out. The research question is actually indicated in the introduction, P2L17. However, the introduction does not lead the reader to this question, nor is a sequence of investigative steps directly developed from it.

3. On P7L7-8 the generation of the necessary 2D wind fields evolving in time from raw met tower measurement data is only indicated but not sufficiently explained. As this is a central step in data processing where assumptions and simplifications have to be applied, a more detailed description is necessary.

4. On P7L31-32 it is explained that simulations were performed using the NREL Wind-PACT 1.5MW WT. Compared to the current state of technology, this is a rather small machine. It seems that at least the NREL 5MW WT would have been more appropriate, especially considering the focus of the manuscript on future, very large and tall WTs. Please comment on this and justify the WT model selection.

5. The results indicate reduced loads and deflection amplitudes for several WT components in case of an LLJ. These improvements should be made quantitative in some meaningful way and presented, e.g., in a table, probably only for selected cases. Thus an impression of their relevance would be given.

Specific comments

1. Fig. 1 could probably be canceled. Its information contents is minimal and mostly repeated in Fig. 2, if the caption is adapted accordingly.

2. Notation: The vertical axes in Figs. 3-10 are labeled $(z-z_0)/R$ while the captions mention "Variation with \xi", and Eq. (3) defines \xi slightly different from the figures. Please clarify and use \xi consistently throughout the manuscript, avoiding confusion of the reader. For the same reason also use the symbol \xi in axis labelling of the figures if appropriate.

3. In the conclusions, P18L6, the term "torque" should be explained more precisely. Torque on which component(s) and with respect to which axis?

---

## Author Comment (AC1) · 15 Sep 2017

**Impacts of the Low-Level Jet's Negative Wind Shear on the Wind Turbine**

Walter Gutierrez, Arquimedes Ruiz-Columbie, Murat Tutkun, and Luciano Castillo

**Response to Referee # 1**

**We are thankful to the referee for his comments that significantly improved the quality of our manuscript. Responses are indicated below and proper changes are highlighted in the new version of the manuscript.**

**Major comments:**

1) **Referee comment:** The literature review is way too long. Also you have a whole section with two subsections dedicated to it besides the introduction where there is also some literature review. My suggestion is that the literature review should be part of the introduction and you need to refer to the important work on the matter (you have many references to non-valuable work). But please but do not make it much longer than it is!!!

   **Author's response:** We highly appreciate the referee for the recommendation, which contribute to make the manuscript more concise.

   **Author's changes:** Sections "Introduction" and "Literature Review has been combined into a single section named "Introduction". The former subsection "Data collection methods" has been renamed "Previous experiences on data collection". Overall, the number of works cited was also significantly reduced. Finally, the last paragraph of the former section "Introduction" has been modified to update the section numbers of the cross-references.

2) **Referee comment:** You use the met tower data as an input for your aeroelastic computations. But from a tower (with different sensors at different vertical levels) you can derive a "2D" wind field. What do you assume to get the 3D wind field or how do you get it anyway?

   **Author's response:** As the reviewer mentioned (specific comment # 26, Page 7 line 8), the tower is a line of measurements which provides one dimension in the vertical axis. The series of collected data generate the second dimension in the streamwise axis. Finally, an assumption must be made to generate the third dimension in the spanwise axis and complete the 3D model. Data from the tower reveal that LLJs are very stable phenomena in terms of wind speed and direction, both of which vary very slowly with time. Moreover, the LLJ's horizontal scales are large, both along and across the wind direction. These two observations support the expectation that horizontal rotational motions are mostly insignificant within LLJs winds and therefore, that the wind speed vector would be quite similar in the spanwise vicinity of the tower. The third dimension in the spanwise axis was thus obtained by replicating the measurements obtained at the tower at the same height.

   **Author's changes:** A paragraph was added immediately thereafter with the above explanation.

3) **Referee comment:** As I mentioned before, I have the opinion that this manuscript reads more like a technical report than a paper for a scientific journal (the only exception is the discussion part that tries to give value to the work). The authors do not state clear how innovative their approach is. It just seems quite standard to perform aerolastic simulations using wind fields with certain characteristics?

   **Author's response:** In our opinion, the most innovative point of this research is the methodical approach to the study of negative wind shear impacts on structures like wind turbines. Up to now, the presence of negative shears inside the turbine's rotor in relation to the presence of positive shears has been largely ignored. A parameter has been proposed to quantify that presence in future studies of LLJs – wind turbines interactions.

   The study demonstrated that the presence of negative shears is significant in reducing the loading on wind turbines. A major conclusion of this study is that the wind turbines of the future should probably be designed with the aim of reaching more often the top of the nightly boundary layer and therefore the altitudes where negative shears are more frequent. Doing so will help to reduce the positive shear's associated damages and to capture the significant LLJ energy.

   The use of aeroelastic simulations of the turbine responses is only a tool within the whole approach. It is worth to notice that rather than using synthetic profiles to generate the wind data, all simulations were based on real data captured at the high frequency of 50 Hz, which allowed to perform the analysis of turbines impact with real-life, small scales of wind motions. On the other hand, the use of the simulator of the turbine responses is dictated by necessity, as it is not practical nor necessary at this point to set up experimental wind turbines with variable heights at the intervals where LLJs are frequent.

   **Author's changes:** Parts of the author's response above have been inserted into the Abstract.

4) **Referee comment:** As mentioned in the discussion the negative shears can have a smaller impact on the loads than the positive shears because the positive shears are larger in magnitude. I think it is important that after describing the met data, pdfs of the wind shear should be shown to see how different these are. Also, what will be the result if the magnitude of negative wind shears was as big as the positive ones? That will be interesting to see in the simulations.

   **Author's response:** The authors appreciate the valuable comments and recommendations by the reviewer. Future research is intended to simulate the turbine's responses with synthetic jets in which the magnitude of the positive and negative shears coincide.

   As the present study has been based on real wind-data, the magnitude of the positive shear below the peak was always greater than the magnitude of the negative shear above. This behavior is expected to occur in the vast majority of real cases. The reason behind this

asymmetry is that the velocity below the peak has to decrease more (from maximum at the peak down to zero at the ground), compared to the velocity above the peak that only need to decrease from maximum at the peak to geostrophic in the free atmosphere.

**Author's changes:** The response to this comment has been inserted into the Discussion and Conclusion section

5) **Referee comment:** During LLJ conditions the variation of the wind shear might be large but so the wind veer and that might have a large(r) impact on the loads. This needs to be addressed in the manuscript.

**Author's response:** In an early step of this research, several of the LLJ's features were correlated with the turbine's responses. Those features included wind speed, wind shear, wind veer and potential temperature. It was found that the feature that correlated more strongly and with more number of turbine's responses was the wind shear, followed closely by the wind speed, then the wind veer and finally the potential temperature.

**Author's changes:** The response above has been inserted into the Introduction.

6) **Referee comment:** Finally the authors repeat too many sentences throughout the manuscript than can be deleted without losing connection or detriment. I will point to some in the specific comments.

**Author's response:** The authors thanks the referee finding those repetitions. Reducing those repetitions will definitely improve the quality of the manuscript.

**Author's changes:** The repetitions were removed, as seen in the response to several specific comments.

**Specific comments:**

1) **Referee comment:** Page 1 line 6 "at the heights of the turbine" For the abstract to stand alone, you need to say at what heights relative to the turbine hub height you refer to here.

**Author's response:** The authors thank the referee for the recommendation.

**Author's changes:** The phrase "at the heights of the turbine" was replaced by "at the heights of the turbine's rotor".

2) **Referee comment:** Page 1 line 9 add "the tower's" before "slower velocities" otherwise the reader might think you are talking about the flow.

**Author's response:** The authors thank the referee for the recommendation.

**Author's changes:** The phrase "slower velocities" was replaced by "the tower's slower velocities".

3) **Referee comment:** Page 1 line 11 "slightly more stable" is quite ambiguous. Perhaps the whole sentence is not needed.

**Author's response:** The authors appreciate the recommendation. In this case, it is of interest to point out that the variance of power is also reduced by the presence of negative shears.

**Author's changes:** The phrase "the power output was slightly more stable" was replaced by "the variance of power production was also reduced".

4) **Referee comment:** Page 1 line 12 replace "the heights of more LLJs" by "LLJ's typical heights".

**Author's response:** The authors thank the referee for the recommendation

**Author's changes:** The phrase was replaced as per referee's recommendation.

5) **Referee comment:** Page 1 lines 15-16 replace "are mainly detected" by "often occur".

**Author's response:** The authors thank the referee for the recommendation.

**Author's changes:** The phrase was replaced as per referee's recommendation.

6) **Referee comment:** Page 1 line 19 shouldn't you move the second "is" after "influence"?

**Author's response:** The authors thank the referee for noticing this typo.

**Author's changes:** The phrase "what is their influence" was replaced by "what their influence is".

7) **Referee comment:** Page 1 line 20 add "turbine" before "structure".

**Author's response:** The authors thank the referee for the recommendation.

**Author's changes:** The word was added as per referee recommendation.

8) **Referee comment:** Page 2 line 6 leave a small space between numbers and units throughout the whole manuscript (this I think it is the first instance).

   **Author's response:** The authors thank the referee for the recommendation.

   **Author's changes:** The space was inserted here and in successive instances.

9) **Referee comment:** Page 2 lines 28-end is not needed + already said. Remember to cleverly, efficiently and shortly combine sections 1 and 2!

   **Author's response:** The authors thank the referee for the recommendation that contribute to make the paper more concise.

   **Author's changes:** The lines pointed out by the referee have been removed and the two sections have been combined (please see response to major concern # 1.)

10) **Referee comment:** Page 3 lidar and sodar can be used already as words so you can save the explanation of the acronyms

    **Author's response:** The authors accept the recommendation.

    **Author's changes:** the text now just refers to "SODAR devices" and "LIDAR devices" without explaining the acronyms.

11) **Referee comment:** Page 3 line 12 "which in wind energy… 1/7" This is not true. If somebody is doing this they are doing it wrong. 1/7 is the shear for a specific roughness, height, and stability conditions (in flat terrain)

    **Author's response:** The authors thank the referee and agree with the observation; however, the point is to show that the IEC standards recommend certain values for normal conditions. Although the shear exponent has been found to depend on parameters such as stability conditions, roughness and height, the IEC standards recommend fixed values for normal conditions. IEC Part I (International Electrotechnical Commission 2005-08) proposes values of $\alpha = 0.2$ for onshore projects, while IEC Part III (International Electrotechnical Commission 2009) specifies $\alpha = 0.14$ for offshore projects.

    **Author's changes:** The phrase has been removed as part of the literature review reduction.

12) **Referee comment:** Page 3 lines 24-25 "could be estimated using an interpolation formula" If this is indeed true then nobody should refer to that paper at all because there are no physics in that

**Author's response:** In fact, this study confirms that there is a relation between the roughness and the shape of the wind shear. Roughness is absent in the IEC recommendations (cited in the response to the previous comment).

13) **Referee comment:** Page 3 line 30 add "a" after "used"

**Author's response:** The authors thank the referee for noticing this issue.

**Author's changes:** The word has been added as per the referee recommendation.

14) **Referee comment:** Page 4 "More recently… jet peak" The power law is not physical so the reference to that study is not needed at all. The strange thing will be that they find the power law to be applicable to LLJs

**Author's response:** The authors agree that the power law cannot fit the LLJ profile, especially due to the presence of the velocity peaks. However; the authors of the cited study were evaluating whether the power law could provide a decent fit just in the region of positive shear well below the peak. If successful, this may have some practical values in estimating some LLJs effects on turbines, as most of the time the LLJ's peak occur well above of most current wind turbines. On the other hand, the authors agree with the referee that the power law does not convey a physical meaning. Moreover, even if the power law were successful in modeling the positive shear zone well below the peak, it would not be helpful for our studies as we investigated the region of negative shears as well.

**Author's changes:** The sentence has been following the referee recommendation.

15) **Referee comment:** Page 4 line 9 the sentence is not needed as it is obvious. The strange thing will be the opposite

**Author's response:** The authors thank the referee for the recommendation.

**Author's changes:** The sentence has been rewritten as follows: " Although expensive, experimental data acquisition methods are better in capturing LLJ information."

16) **Referee comment:** Page 4 line 19 you mean "wavelet" instead of "wavelength"?

**Author's response:** The authors accept the recommendation.

**Author's changes:** The word has been changed.

17) **Referee comment:** Page 4 lines 21-22 this sort of conclusion is not true. You refer to some literature regarding measurements with remote sensors that are not valuable for comparison. There are many papers where winds (and LLJs) have been observed with lidars (for example) and that you are not referring to

**Author's response:** The authors accept that some rephrasing is helpful. What we want to highlight here is that, even though devices like lidars can detect the presence and features of structures such as LLJs, their time resolutions are generally coarser and therefore some important frequencies are filtered out. By contrast, 20- or 50- Hz instruments installed on meteorological towers can capture important frequencies that can affect the wind turbine.

**Author's changes:** The sentences are restated as follows: "In summary, high-frequency instruments installed on meteorological towers are the best option to capture the scales of wind motions that structurally affect wind turbines. Although devices like LIDARs can detect the presence and features of structures such as LLJs, their time resolutions are generally coarser and therefore some important frequencies are filtered out."

18) **Referee comment:** Page 4 line 29 "AEP at night" I guess you mean production AEP is in a yearly basis and is one single number

**Author's response:** The authors thank the referee for bringing this statement for clarification. Instead of the usual parameter AEP obtained by calculating the total energy produced during the year, the authors of the cited study performed the summation of the energy produced in the year during the nightly hours and compared the result to the annual sum of energy during the daily hours. So basically, they compared the Annual Energy Produced during the night with the Annual Energy Produced during the day.

**Author's changes:** To avoid confusion, the word "AEP" has been replaced by "energy production".

19) **Referee comment:** Page 4 line 30 "to the wind farm" which wind farm? You are referring to some work from Antoniou which is not that much relevant. The danes have made many other campaigns with lidar measuring the wind profile all the way up to the ABL height that are relevant for your study

**Author's response:** The wind farm is the "Danish Test Station for Large Wind Turbines" mentioned at the end of the sentence. Because our focus is on LLJs incidents whose peak occurred within the heights of the turbine's rotor, we skipped the literature of works that, although interesting, only concentrated on the meteorological side of the LLJs and preferred instead those studies whose main goal was to analyze their influence on wind turbines. Nevertheless, the authors follow the recommendation of removing the citation as other papers mentioned conveys the same message.

**Author's changes:** Both citations have been removed.

20) **Referee comment:** Page 5 line 15 you mean "farm" instead of "turbine"?

   **Author's response:** The authors thanks the referee for noticing this issue.

   **Author's changes:** The word has been replaced.

21) **Referee comment:** Page 6 line 9 replace "explain" by "explains"

   **Author's response:** The authors thanks the referee for noticing this issue.

   **Author's changes:** The tense has been modified.

22) **Referee comment:** Page 6 line 8 and 12 replace "200-meter's" by "200-m"

   **Author's response:** The authors thanks the referee for the recommendation.

   **Author's changes:** The change was applied at all instances in the document.

23) **Referee comment:** Page 6 line 27 where in Stull (1988)?

   **Author's response:** The authors thanks the referee for the recommendation.

   **Author's changes:** The page number was added to the citation.

24) **Referee comment:** Page 6 lines 29-30 "The data… queries" is not needed. Also how you know they are "corrupted" or "inaccurate"? "the conversion to a normalized data structure… paremeters" is not needed either

   **Author's response:** There are several conditions that were checked to ensure the reliability of the data, which can be considered standard data management procedures. E.g. values that make no sense for a parameter or with a "N/A" (not available), blank measurements at some height or for specific variables, etc.

   **Author's changes:** Following the recommendation from the reviewer, the three sentences were reduced to "The information was periodically transferred into a central database which was scanned in search…".

25) **Referee comment:** Page 7 line 4 what do you mean by "sustained winds" and by "preference" (you use them or not)?

**Author's response:** The authors thanks the referee for requesting to clarify those phrases.

**Author's changes:** The phrase "sustained winds" was replaced by "wind speed above a threshold". The phrase "Preference was then given to LLJs cases whose peak…" has been replaced by "LLJs cases selected were those whose peak…".

26) **Referee comment:** Page 7 line 8 "resolution of 1 m in both directions" which 2? It is just tower data! If you meant the turbulence box then you need to say how you go from a tower to a turbulence box where you need to model the three wind speed components. What assumptions you made for coherence?

**Author's response:** As the reviewer mentioned, the tower is a line of measurements which provides one dimension in the vertical axis. The series of collected data generate the second dimension in the streamwise axis.  Finally, an assumption must be made to generate the third dimension in the spanwise axis and complete the 3D model. Data from the tower reveal that LLJs are very stable phenomena in terms of wind speed and direction, both of which vary very slowly with time. Moreover, the LLJ's horizontal scales are large, both along and across the wind direction. These two observations support the expectation that horizontal rotational motions are mostly insignificant within LLJs winds and therefore, that the wind speed vector would be quite similar in the spanwise vicinity of the tower. The third dimension in the spanwise axis was thus obtained by replicating the measurements obtained at the tower at the same height.

**Author's changes:** A paragraph was added immediately thereafter with the above explanation.

27) **Referee comment:** Page 7 lines 10 -11 Not needed, already said

**Author's response:** The authors thanks the referee for the recommendation.

**Author's changes:** The sentence was omitted.

28) **Referee comment:** Page 7 line 15 replace "height above the ground level of the turbine height" by "turbine hub height"

**Author's response:** The authors thanks the referee for the recommendation.

**Author's changes:** The recommendation was implemented.

29) **Referee comment:** Page 7 lines 25-27 Not needed, already said

**Author's response:** The authors thanks the referee for the recommendation.

**Author's changes:** The sentence was erased.

30) **Referee comment:** Fig. 2 The axes in your figures cannot be read. You need to increase the size of the labels of nearly all figures

**Author's changes:** The size of all figures has been augmented.

31) **Referee comment:** Page 8 After "Results" this is repeated all over the manuscript. Delete!

**Author's response:** The authors thanks the referee for the recommendation.

**Author's changes:** The first sentence was deleted.

32) **Referee comment:** Page 9 line 2 replace "by looking at plots of" by "with"

**Author's response:** The authors thanks the referee for the recommendation.

**Author's changes:** The proposed change was implemented.

33) **Referee comment:** Page 9 lines 24-31 this information is not needed, is too elemental about pdfs and plotting. Also in figure 3 frames b and c contain the same information. Cannot read the axes labels!

**Author's response:** In fact, this part was added to the manuscript after feedback that recommended to explain the genesis of the pdf plots in order to make them more accessible to a broader audience. However, posterior recommendations, including the reviewer's one, pointed out that the plots are common knowledge and therefore this explanation can be omitted. After careful consideration, it has been concluded that the next paragraph is sufficient and therefore the reviewer recommendation has been followed.

**Author's changes:** Figure 3 and the paragraphs referenced by the speaker have been removed.

34) **Referee comment:** Page 10 all the text is not needed and/or has been said before. The description of what the lines are should be in the caption

**Author's response:** The authors thanks the referee for the recommendation.

**Author's changes:** The remaining descriptions have been moved to the figures' caption.

.

35) **Referee comment:** Page 11 lines 2-7 This can be omitted

   **Author's response:** The authors thanks the referee for the recommendation.

   **Author's changes:** The paragraph has been removed.

36) **Referee comment:** Page 11 lines 8-9 Is this a statement or an observation from your analysis?

   **Author's response:** This is a key observation from a previous study.

   **Author's changes:** A citation to the study has been added.

37) **Referee comment:** Page 11 line 13 "are very skewed" where? I cannot see skewness

   **Author's response:** In fact, the figure at right shows a very-skewed distribution for all $\xi$. Different from a symmetric distribution, a skewed distribution is often identified in a 2D-plot by the presence of a long tail at one side and a big peak at the opposite site, i.e. the center of the distribution is shifted from the geometric center. In the plots shown, the PDF values are represented by the color depth and peaks can be detected where the color is darker. The figure at right shows that the distribution for the radial component has the big peaks (visualized by the dark tones) completely shifted to the left for all values of $\xi$, which evidence that the distribution is very skewed to the right. A more revealing evidence of skewness is the long tail at the right side, identified by the longer distance between peak and percentile 97.5, compared to the distance between peak and percentile 2.5. Finally, the skewness is also disclosed by the separation between mean and median values.

38) **Referee comment:** Page 11 line 16 change "conclusion" by "observation"

   **Author's response:** The authors thanks the referee for the recommendation.

   **Author's changes:** The word has been changed as recommended.

39) **Referee comment:** Page 12 line 5. Similar to comment 38

   **Author's response:** The authors thanks the referee for the recommendation.

   **Author's changes:** The word has been changed as recommended.

40) **Referee comment:** Page 12 lines 12-13 Not needed, delete please

   **Author's response:** The authors thanks the referee for the recommendation.

   **Author's changes:** After careful consideration, the sentence was deleted.

41) **Referee comment:** Page 12 lines 18-19 similar to comment 40

   **Author's response:** The authors thanks the referee for the recommendation.

   **Author's changes:** The first sentence was deleted.

42) **Referee comment:** Page 12 line 23 similar to comment 38

   **Author's response:** The authors thanks the referee for the recommendation.

   **Author's changes:** The word has been changed as recommended.

**References**

Gutierrez, Walter, Guillermo Araya, Praju Kiliyanpilakkil, Arquimedes Ruiz-Columbie, Murat Tutkun, and Luciano Castillo. 2016. "Structural impact assessment of low level jets over wind turbines." *Journal of Renewable and Sustainable Energy* 8 (2). doi:http://dx.doi.org/10.1063/1.4945359.

International Electrotechnical Commission. 2005-08. "Wind Turbines - Part 1: Design requirements." In *International Standards*, 1-92. Geneva: International Electrotechnical Commission.

International Electrotechnical Commission. 2009. "Wind Turbines - Part 3: Design requirements for offshore wind turbines." In *International Standards*. International Electrotechnical Commission.

---

## Author Comment (AC2) · 15 Sep 2017

**Impacts of the Low-Level Jet's Negative Wind Shear on the Wind Turbine**

Walter Gutierrez, Arquimedes Ruiz-Columbie, Murat Tutkun, and Luciano Castillo

Response to Referee # 2

**We are thankful to the referee for his comments that significantly improved the quality of our manuscript. Responses are indicated below and proper changes are highlighted in the new version of the manuscript.**

**Major comments:**

1) **Referee comment:** Alltogether, the manuscript is lengthy. The authors should consider to identify the key cases and results and concentrate on these. Also the literature review, while much appreciated, appears to be more extensive than necessary.

   **Author's response:** We highly appreciate the referee for the recommendation, which contribute to make the manuscript more concise.

   **Autor's changes:** "Literature Review" has been merged with "Introduction" into a single section named "Introduction". The former subsection "Data collection methods" has been renamed "Previous experiences on data collection". Overall, the number of references has been significantly reduced. Finally, the last paragraph of the former section "Introduction" has been modified to update the section numbers of the cross-references. On the other hand, the explanation of the PDF (in former section 4.1) has been compacted.

2) **Referee comment:** The manuscript reads quite technical and should be improved in terms of a dedicated research question and a rigorous sequence of steps and arguments to give answers to that question. Most of the necessary elements are already mentioned in the paper, but they should be better worked out. The research question is actually indicated in the introduction, P2L17. However, the introduction does not lead the reader to this question, nor is a sequence of investigative steps directly developed from it.

   **Author's response:** The authors thanks the referee for the recommendation, which undoubtedly increase the interest on the manuscript.

   **Autor's changes:** The final paragraphs of the "Introduction" have been modified to highlight the importance of the research question. Two paragraphs are now devoted to the reasoning leading to the research question, from " In some parts of the world such as Europe, wind turbines…" to "… or mitigates those effects."

   Thereafter, the sequence of investigative steps has been introduced as follows: " To answer this question, a process has been devised to find out how the turbines responses vary with the presence of negative wind shears. First, enough wind information was collected to allow the quest of typical LLJ's incidents. Second, a parameter was devised as independent variable that quantify the proportion of rotor area that receives negative shears. Then, cases

were generated by gradually modifying the parameter. Finally, simulations of the turbine responses were performed for each case and the results were compared to draw conclusions."

3) **Referee comment:** On P7L7-8 the generation of the necessary 2D wind fields evolving in time from raw met tower measurement data is only indicated but not sufficiently explained. As this is a central step in data processing where assumptions and simplifications have to be applied, a more detailed description is necessary.

**Author's response:** The authors thanks the reviewer for this insightful observation that clearly definitely makes clearer to the reader the procedures used.

**Autor's changes:** The following text has been added immediately thereafter: "One decision to make was how to construct a 3D box of wind speed information. The tower is a line of measurements which provides one dimension in the vertical axis. The series of collected data generate the second dimension in the streamwise axis. Finally, an assumption must be made to generate the third dimension in the spanwise axis and complete the 3D model. Data from the tower reveal that LLJs are very stable phenomena in terms of wind speed and direction, both of which vary very slowly with time. Moreover, the LLJ's horizontal scales are large, both along and across the wind direction. These two observations support the expectation that horizontal rotational motions are mostly insignificant within LLJs winds and therefore, that the wind speed vector would be quite similar in the spanwise vicinity of the tower. The third dimension in the spanwise axis was thus obtained by replicating the measurements obtained at the tower at the same height."

4) **Referee comment:** On P7L31-32 it is explained that simulations were performed using the NREL Wind- PACT 1.5MW WT. Compared to the current state of technology, this is a rather small machine. It seems that at least the NREL 5MW WT would have been more appropriate, especially considering the focus of the manuscript on future, very large and tall WTs. Please comment on this and justify the WT model selection.

**Author's response:** The authors agree that there exists a loose trend of having larger wind turbines at greater altitudes. The tendency nonetheless not always holds; for example, the current world's tallest wind turbine is a Nordex 3.3 MW that reach 230 m but has only a capacity of 3.3 MW. The turbine surpassed the height of the previous tallest one, the Vestas V164, by around 10 m, even as the latest is more than 2.5 times as powerful (8 MW).

For our simulations, we needed to select a single wind turbine that could fit a broad range of altitudes, from less than 100 m to almost 200 m (measured from the ground up to the height of the tip of the blade in the upper position). The reason is that we have been studying and comparing the impacts of the LLJ in many situations, starting with the peak of the jet occurring much higher than the turbine rotor and ending with the jet peak impacting below the rotor. We needed to ensure that the turbine model was the same to ensure a fair comparison among the cases.

To select a wind turbine model, not only the hub height was important, but also the rotor radius, in order to keep the rotor within the altitude constraints for all simulations. First, a blade tip in the upper position should never reach 200 m (the limit of the wind measurements from the tower) even when testing cases with the jet peak impacting below the turbine rotor. Second, a blade tip in the lower position should never approach the ground even when testing cases with the jet peak impacting above the turbine rotor. With a rotor diameter of just 70 m, the NREL Wind- PACT 1.5MW WT is perfect to fit in all simulation cases. As a contrast, the NREL Wind- PACT 5MW WT has a rotor radius of 128 m, which makes almost impossible to comply with the constraints for all cases.

5) **Referee comment:** The results indicate reduced loads and deflection amplitudes for several WT components in case of an LLJ. These improvements should be made quantitative in some meaningful way and presented, e.g., in a table, probably only for selected cases. Thus an impression of their relevance would be given.

**Author's response:** The authors thanks the reviewer for this recommendation. Careful consideration was given as to how summarize results in a table. However, there is no simple parameter that can synthetize the information in the graphs. For example, several statistical parameters are visualized directly or indirectly: means, medians, percentiles, skewness, etc. and each of them only tells part of the story. As in many cases, the graphs are compelling in putting together all massive information to make evident to the reader in a glimpse the salient points.

The information is shown in terms of variations with $\xi$. Consolidation of variables (e.g. the maximum of the medians, etc.) may be attempted; however, they would provide limited information compared to visualizing the trends across all values of $\xi$.

**Specific comments:**

1) **Referee comment:** Fig. 1 could probably be canceled. Its information contents is minimal and mostly repeated in Fig. 2, if the caption is adapted accordingly.

**Author's response:** The authors thank the referee for the recommendation.

**Autor's changes:** Figure 1 has been removed and the following text has been added immediately after the definition of $\xi$: "The parameter $\xi$ is a continuous variable. Some characteristics values are of interest. First, if $\xi = 1$ then the peak of the jet impacted exactly at the altitude of the lowest point of the turbine's sweep area and thus the wind shear was entirely negative across the turbine's rotor. Second, if $\xi = 0$ then the peak of the jet occurred exactly at the height of the turbine's hub and thus the wind shear was positive below the hub and negative above. Finally, if $\xi = 1$ then the peak of the jet impacted exactly at the altitude of the highest point of the turbine's sweep area and thus the wind shear was entirely positive across the turbine's rotor."

2)  **Referee comment:** Notation: The vertical axes in Figs. 3-10 are labeled (z-z_0)/R while the captions mention "Variation with nxi", and Eq. (3) defines nxi slightly different from the figures. Please clarify and use nxi consistently throughout the manuscript, avoiding confusion of the reader. For the same reason also use the symbol nxi in axis labelling of the figures if appropriate.

   **Author's response:** The authors thank the referee for noticing the mismatch.

   **Autor's changes:** The y-axes labels have been changed to the variable name.

3)  **Referee comment:** In the conclusions, P18L6, the term "torque" should be explained more precisely. Torque on which component(s) and with respect to which axis.

   **Author's response:** The authors thanks the reviewer for the recommendation to clarify the term.

   **Autor's changes:** The sentence has been rephrased as follows: "Therefore, the magnitude of the torque created by the negative shear upon the long elements (such as the blades and the tower), especially around the spanwise axis, becomes smaller than the one created by the positive shear."